# Satellite-Based Estimation of Daily Ground-Level PM$_{2.5}$ Concentrations over Urban Agglomeration of Chengdu Plain

**Weihong Han** and **Ling Tong** *

School of Automation Engineering, University of Electronic Science and Technology of China, Chengdu 611731, China; weihong_han@std.uestc.edu.cn
* Correspondence: tongling@uestc.edu.cn; Tel.: +86-028-61831792

**Abstract:** Monitoring particulate matter with aerodynamic diameters of less than 2.5 μm (PM$_{2.5}$) is of great importance to assess its adverse effects on human health, especially densely populated regions. In this paper, an improved linear mixed effect model (LMEM) was developed. The model introduced meteorological variable, column water vapor (CWV), which has as the same resolution as satellite-derived aerosol optical thickness (AOT), to enhance PM$_{2.5}$ estimation accuracy by considering spatiotemporal consistency of CWV and AOT. The model was implemented to urban agglomeration of Chengdu Plain during 2015. The results show that model accuracy has been improved significantly compared to linear regression model (R$^2$ = 0.49), with R$^2$ of 0.81 and root mean squared prediction error (RMSPE) of 15.47 μg/m$^3$, mean prediction error (MPE) of 11.09 μg/m$^3$, and effectively revealed the characteristics of spatiotemporal variations PM$_{2.5}$ level across the study area: The PM$_{2.5}$ level is higher in the central and southern areas with dense population, while it is lower in the northwest and southwest mountain areas; and the PM$_{2.5}$ level is higher during autumn and winter, while it is lower during spring and summer. The product data in this paper are valuable for local government pollution monitoring, public health research, and urban air quality control.

**Keywords:** urban pollution; remote sensing; PM$_{2.5}$; AOT

## 1. Introduction

PM$_{2.5}$, particulate matter with aerodynamic diameters of less than 2.5 μm, is a significant indicator of air pollution level, which has aroused the attention of environmental protection authorities from central and local government in China. Numerous epidemiological studies have demonstrated that there are strong association between PM$_{2.5}$ exposure and public illness and mortality from respiratory and cardiovascular diseases [1–9]. Ground-based monitoring sites can measure PM$_{2.5}$ level continuously in the long term, but the monitoring sites are generally distributed sparsely and unevenly due to geological and environmental condition restriction, accordingly causing dense distribution in urban area and spare in rural area. Ground-based monitoring networks are inadequate to analyze fine-scale spatial variability of pollution, which is important for health impact assessment and needed to characterize the heterogeneity of human exposure to PM$_{2.5}$, and effectively prevent and control air pollution [10]. There are many publications that highlight the problem of sparse PM$_{2.5}$ locations. Let us start with the review paper of Hoff and Christopher 2009 [11].

However, aerosol optical thickness (AOT), one parameter of the characteristics of aerosol, is equal to the integral extinction coefficient over a vertical column of atmosphere at observation location, and it represents the degree to which aerosol prevent the transmission of light by absorption or scattering of light. AOT can be derived from satellite imageries indirectly [12–15], and numerous global satellites are

widely used for AOT retrieval, e.g., advanced very high-resolution radiometer (AVHRR), total ozone mapping spectrometer (TOMS), moderate-resolution imaging spectroradiometer (MODIS), multi-angle imaging spectroradiometer (MISR), sea-viewing wide field-of-view sensor (SeaWifFS) [16], and visible infrared imaging radiometer suite (VIIRS) [17], continuously measuring AOT from the late 1990s to the most recent.

Satellite data are able to monitor large/urban scale air quality through adding synoptic and spatial distribution information to ground-based air quality measurements and modeling [18,19]. In recent decades, numerous researchers have developed many algorithms for calibrating satellite-derived AOT to ground-level $PM_{2.5}$ concentrations. Early studies primarily established simple linear regression models to relate $PM_{2.5}$ and AOT [19–21]. Then, meteorological data were incorporated in the multiple regression analysis and improved the $PM_{2.5}$–AOT correlation [22]; advanced statistical models, such as artificial neural networks (ARN) [23], generalized additive model (GAM) [24,25], land use regression model (LUR) [26–28], and geographically weighted regression (GWR) [29–33] were developed to analyze the variability of $PM_{2.5}$–AOT relationship correlated with meteorology or land use data. Although such advanced models gained higher estimation accuracy for large regions such as the northeastern and southeastern US, Europe, and China, etc., auxiliary parameters increased the complexity of these models and introduced other errors. Lee firstly incorporated time-varying parameters (day-to-day) in a statistical model to calibrate MODIS AOT and obtained good estimation accuracy [34]; and this model was further improved through adding meteorological data, land use data, and pollution sources over the whole China, Beijing, Beijing–Tianjin–Hebei (BTH) region, Yangtze River Delta (YRD), Peral River Delta (PRD), California, and southeastern U.S. [33,35–45].

In the presence of water in the atmosphere, an aerosol particle changes its size and other characteristics by absorbing or evaporating water (i.e., aerosol hygroscopic growth) [46]. Because AOT is an indicator of the abundance of aerosol particles in the total vertical column while $PM_{2.5}$ concentration is measured at ground level, the correlation between them is influenced by the vertical distribution of aerosols and the humidity that impacts aerosol extinction coefficient. These two factors are associated with atmospheric profiles, ambient conditions, as well as the size distributions and aerosol particles components, all of which may have large spatial and temporal variations. Hence, $PM_{2.5}$ estimates from AOT alone will have large uncertainties. To reduce these uncertainties, numerous studies have incorporated the planetary boundary layer height, ambient relative humidity, and other meteorological factors (e.g., temperature, wind speed) in their models to better characterize the correlation between AOT and $PM_{2.5}$ [21,22,24,29,32,33,37–39,42,47–49]. However, it is usually difficult to obtain those observational parameters, such as particle vertical distribution at specific locations. Most of the works mentioned above were conducted using data obtained from the sparse ground-based monitoring sites [33,38,47,48] and model assimilated meteorological data, e.g., Goddard earth observation system data assimilation system [21,39,42], rapid update cycle [22,24] North American regional reanalysis [29], European Center for Medium-Range Weather Forecasts [32], and North American land data assimilation system [37]. Because of the establishing strategies of ground-based monitoring sites, there are very few, even no, monitoring sites in exurban and rural areas, small towns, etc., which may cause some uncertainties in model retrieval. Meanwhile, model-assimilated meteorological data can have better spatial coverage over all the locations, but their grid spatial resolution is low such as 0.25° latitude × 0.3125° longitude and 13 km, which are much lower than that of satellite AOT, causing serious inconsistence of spatial coverage between meteorological data and AOT grid cell. As discussed above, many meteorological variables can substantially affect the $PM_{2.5}$–AOT relationship and are significant predicators of $PM_{2.5}$ concentrations, and the meteorological data source also introduces some uncertainties.

Conventional Collection 5.1 (C5.1) MODIS AOT at 10 km spatial resolution and Collection 6.0 (C6) MODI AOT at 3 km spatial resolution were widely used to estimate ground-level $PM_{2.5}$ concentration due to its daily global coverage and availability in the long term. Nevertheless, those spatial resolutions are not enough to capture the spatial variation of air pollution in urban scale [50,51].The multi-angle

implementation of atmospheric correction (MAIAC) algorithm were developed for the MODIS aerosol retrievals and atmospheric correction over both dark vegetated surfaces and bright deserts based on a time series analysis and image-based process, producing products such as AOT, spectral regression coefficient (SRC), column water vapor (CWV), and bidirectional reflectance distribution function (BRDF) at 1 km spatial resolution, which is finer than previous MODIS AOT products [52,53]. More importantly, the MAIAC CWV is generated along with MAIAC AOT from MAIAC algorithm, so both of them have the same spatial coverage and time taken.

In this study, in order to reduce the uncertainty caused by inconsistency of spatiotemporal between AOT and meteorological data, we firstly incorporated MAIAC CWV into $PM_{2.5}$ estimation model based on widely used LMEM method [34]; and the model is implemented to urban agglomeration of Chengdu Plain, which is one of the most heavily polluted in China in 2015 [54–56]; then, we assessed model performance via different ground measurements of $PM_{2.5}$ datasets and MAIAC AOT datasets; the spatiotemporal change of $PM_{2.5}$ level in urban agglomeration of Chengdu Plain was also analyzed. A final table with acronyms is attached at the end of the manuscript, please see Appendix A.

## 2. Materials and Methods

### 2.1. Study Area

The urban agglomeration of Chengdu Plain consists of 8 province-controlled cities: Chengdu, Deyang, Mianyang, Meishan, Ziyang, Suining, and part of Ya'an and Leshan, with a population of around 50 million. It is located in the western part of the Sichuan Basin surrounded by mountains: The Longmen Mountains lie in the northwest, the Qionglai Mountains lie west, and the Longquan Mountains lie east (Figure 1b). Due to the surrounding mountains and temperature inversion, it often experiences foggy weather and smog. The climate has distinct seasons but is moist and cloudy, with the least days of sunshine in China. In recent years, it has undergone heavy air pollution because of the enormous economic boom, urban construction, the increase in the number of motorized vehicles, population growth, surrounding topography, and seasonal weather conditions.

### 2.2. Ground-Level PM2.5 Datasets

Ground-level $PM_{2.5}$ datasets are used for model fitting and model testing in our study. $PM_{2.5}$ data are collected hourly from 36 monitoring sites of national monitoring networks in urban agglomeration of Chengdu Plain (Figure 1c and see Supplementary: Table S1). The ground-based $PM_{2.5}$ monitoring sites are deployed densely in urban area and sparsely in rural area and measured by the tapered element oscillating microbalance method (TEOM) [49,57], which has an accuracy of $\pm 1.5$ μg/m$^3$ for 1 h average, and performs a calibrating process and quality control according to the Chinese National Ambient Air Quality Standard (China NAAQS) [58]. The TEOM continuously pumps in the ambient air and heats it up (to remove the water vapor's influence), and the particles with sizes smaller than 2.5 μm are selected by passing through a special head and then weighted. Thus, the dry mass of $PM_{2.5}$ within a unit volume of ambient air is acquired and recorded every 15 min. It should be noted that by heating the air sample, the measured PM mass (dry) might be less than the real PM level in the ambient air due to the volatilization of semi-volatile aerosol components [59].

In order to better explore the $PM_{2.5}$–AOT relationship, we selected four $PM_{2.5}$ datasets, which are mainly based on two satellites (i.e., Terra and Aqua) overpass. AOTs are measured by Terra satellite (at ~10:30 local time) in the morning and Aqua satellite (at ~13:30 local time) in the afternoon. The four $PM_{2.5}$ datasets are averaged from hourly ground-level $PM_{2.5}$ at different time period: 1) $PM_{2.5}$ 24-h average; 2) $PM_{2.5}$ time average of 10:00 and 14:00; 3) $PM_{2.5}$ time average of 10:00 and 11:00; 4) $PM_{2.5}$ time average of between 13:00 and 14:00 at each monitoring site in 2015.

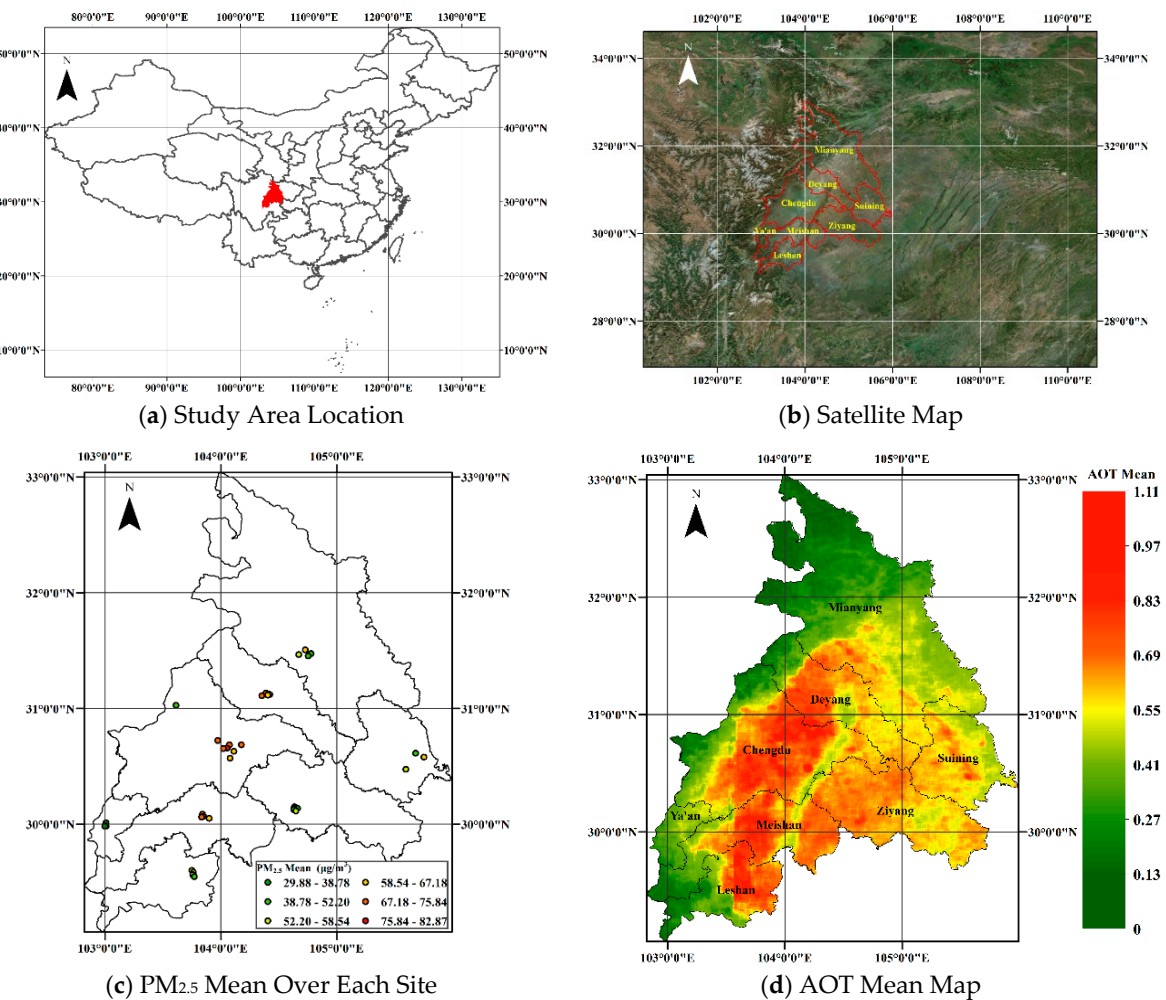

(**a**) Study Area Location          (**b**) Satellite Map

(**c**) PM2.5 Mean Over Each Site          (**d**) AOT Mean Map

**Figure 1.** (**a**) Study area location at China (displayed with red polygon in the map); (**b**) satellite map shows that study area (red lines) is surrounded by mountains in the west, north, and northeast (Satellite imagery source: Esri, DigitalGlobe, GeoEye); (**c**) annual average $PM_{2.5}$ 24-h average ($\mu g/m^3$) for each site, which is classified into six levels represented by different color; (**d**) the spatial distribution of average of multi-angle implementation of atmospheric correction (MAIAC) aerosol optical thickness (AOT) at 1 km resolution during 2015. The boundaries of eight province-controlled cities are shown in black lines.

*2.3. MAIAC Product Datasets*

In our study, the MAIAC product was downloaded for both Terra and Aqua from National Aeronautics and Space Administration (NASA) Center for Climate Simulation via https portal [60]. In order to maintain reasonable file size, the MAIAC algorithm team divided the entire China up into a title grid, and the title grid coordinate system starts at (0, 0) (horizontal tile number, vertical tile number) in the upper left corner and proceeds right (horizontal) and downward (vertical). The study area consists of two tiles, i.e., h02v02 and h03v02. The two tiles include MAIAC data derived from both Terra satellite in the morning and Aqua satellite in the afternoon. The MAIAC algorithm was specifically developed for MODIS data, performing aerosol retrievals and atmospheric correction over both vegetated surfaces and bright deserts at 1 km spatial resolution based on time-series analysis and image-based processing [52,53]. MAIAC algorithm generates products such as aerosol parameters, CWV, BRDF, and SRC. The aerosol parameters include AOT, fine mode fraction, Angstrom exponent from wavelength of 0.47 to 0.67 um, and aerosol type including background, smoke, and dust models [53]. Firstly, MAIAC collected an accumulation of MODIS observations during 5 (over poles)–16

(over equator) with a sliding window approach at specific region. Then, those MODIS data were gridded to 1 km spatial resolution with specific projection coordinate system. When the surface remains stable during collection period, the surface BRDF is retrieved using the regional background aerosol model. Because the 2.1 μm band is transparent to aerosol, it is used to retrieve aerosol and surface reflectance over dark and moderately bright surfaces. Four or more days with low AOT values are used to calculate SRC correlating surface reflectance in the blue and shortwave infrared bands. Once SRC is obtained, the AOT is retrieved with the last MODIS observation. AOT retrieval is conducted with the regional background aerosol model in clear conditions, and surface BRDF in haze conditions. Aerosol robotic network (AERONET) [61] validation shows that the MAIAC and MOD04 algorithms have similar accuracy over dark and vegetated surfaces and have improved the accuracy over brighter surfaces due to SRC retrieval and BRDF factor characterization, as demonstrated for several U.S. west coast AERONET sites [53].

Due to cloudy and foggy weather conditions in the study area, there are many days or areas without satellite AOT measurements. In order to improve spatial coverage of satellite-derived AOT, we merged MAIAC AOT measurements of the Terra satellite (at ~10:30 local time) in the morning and Aqua satellite (at ~13:30 local time) in the afternoon [35,42]. With regard to a given location in one day, there are two situations for AOT availability, one in which both MAIAC Aqua AOT and MAIAC Terra AOT are available, and the other in which just one of MAIAC Aqua AOT and MAIAC Terra AOT is available. If the two AOT measurements (i.e., Terra and Aqua) have strong correlation, we can estimate missing AOT measurement form each other. Hence, in our study, we developed a simple linear regression model with MAIAC Terra AOTs and MAIAC Aqua AOTs in the first situation, then estimate missing AOT measurements from another available one in the second situation. Since the ratio of AOT in the morning to that in the afternoon varied by season, we classified one year into two seasons: Warm season (March to August) and cold season (September to February). The linear regression models of estimating missing AOTs in different season in our study were given as below:

Warm season:

$$\tau'_{Terra} = 0.899 * \tau_{Aqaua} + 0.080 \tag{1}$$

$$\tau'_{Aqua} = 0.885 * \tau_{Terra} + 0.028 \tag{2}$$

Cold season:

$$\tau'_{Terra} = 1.025 * \tau_{Aqua} + 0.046 \tag{3}$$

$$\tau'_{Aqua} = 0.869 * \tau_{Terra} + 0.025 \tag{4}$$

where $\tau$ is the MAIAC AOT value. The $R^2$ of regression models on warm season and cold season were 0.80 and 0.89, respectively. The missing AOT value was estimated using the above regression equation and then the daily AOT was calculated by averaging of MAIAC Terra AOT and MAIAC Aqua AOT. Finally, the daily AOT represents the average aerosol level between 10:00 and 14:00 for each day.

We also evaluated the MAIAC AOT product. Because there is no AERONET site in Sichuan Basin, the ground-based AOT measurements from three AERONET sites in Beijing (Beijing [116.38° E, 39.97° N], Beijing_CAMS [116.31° E, 39.93° N], and Beijing_RADI [116.37° E, 40.00° N]) in the year of 2015 were used to validate the MAIAC AOT. To be comparable with the spectrum setting of MODIS, AERONET AOT at 550 nm were gained by calculating the Angstrom turbidity coefficient and Angstrom exponent with AEROENT measured AOTs at two wavelengths (440 nm and 675 nm) [62,63]. The total number of AOT pairs (MAIAC AOT–AERONET AOT) for the three sites are 257, 272, and 90, respectively. MAIAC AOT shows high temporal correlation with AERONET AOT (see Supplementary: Figure S1), with Pearson correlation coefficients R of 0.96, 0.97, and 0.97 at three sites, respectively, and the mean AOT difference MAIAC and AERONET at three sites is 0.097.

### 2.4. Population Datasets

The gridded population of the world (GPW) collection in fourth version (GPWv4) generated by socioeconomic data and application center (SEDAC) was used in our study. It can provide a spatially disaggregated population layer that is compatible with datasets from social, economic, and remote sensing. GPWv4 used the results of the 2010 round population and housing censuses with much finer spatial resolution. It produces the population distribution for the years 2000, 2005, 2010, 2015, and 2020 [64]. In this study, GPW data in 2015 at 1 km spatial resolution were employed. In our study, min-max normalization process was also performed to change the range of GPW data ranging from 0 to 1.

### 2.5. Data Pre-Processing and Matching

To establish the $PM_{2.5}$-AOT relationship model (parameterization), the ground-based daily $PM_{2.5}$ measurements, gridded population data (POP), satellite-derived MAIAC AOT, and CWV values must be collected at the closest spatial location and time. For each monitoring site ($PM_{2.5}$), the 1 km pixel of MAIAC AOT, CWV, and POP where the monitoring site location is selected. The $PM_{2.5}$, AOT, CWV, and POP is picked to form a sample $PM_{2.5}$-AOT pair. Data pairs with either $PM_{2.5}$ values less than 3.0 μg/m$^3$ or missing $PM_{2.5}$/AOT measurements are omitted. In addition, because the regression slope cannot be estimated from only one single data pair, the days with less than two $PM_{2.5}$-AOT pairs available are excluded.

### 2.6. LMEM Model Fitting and Validation

Lee et al. [34] established day-specific $PM_{2.5}$–AOT relationships using LMEM algorithm and estimated daily estimation of ground-level $PM_{2.5}$ concentrations based on MOD04 AOT in the New England region, considering the day-to-day variability of $PM_{2.5}$–AOT relationship correlated to mixing height, relative humidity, $PM_{2.5}$ composition, and $PM_{2.5}$ vertical profile with the assumption that $PM_{2.5}$–AOT relationship varies largely day to day but minimally spatially on a given day in study area. The LMEM calculates day-specific random intercepts and slopes for the $PM_{2.5}$–AOT relationship and incorporates both fixed-effects terms and random-effects terms. As discussed above, the correlation between $PM_{2.5}$ and AOT is influenced by the humidity that affects aerosol extinction coefficient. Because the aerosol vertical profile during satellite (Terra and Aqua) overpass time in each day is difficult to obtain, the model did not take it into account. Moreover, anthropogenic factors influencing the $PM_{2.5}$–AOT relationship usually are constant, e.g., major road, factory emissions. Hence, we incorporated population data as another indicator of influencing the spatial variability of the $PM_{2.5}$–AOT relationship. It is should be noted that to reduce the uncertainty caused by inconsistency of spatiotemporal between AOT and meteorological data, both of them are should be generated from the same satellite. The improved LMEM equation is below:

$$PM_{ij} = (\alpha + \mu_j) + (\beta + \upsilon_j) \times AOT_{ij} + (\gamma + \varphi_j) \times CWV_{ij} + POP_i + \varepsilon_{ij} \tag{5}$$

where, $PM_{ij}$ is the daily average $PM_{2.5}$ concentration (μg/m$^3$) at site i on day j; $AOT_{ij}$ is the daily average AOT value (unitless) corresponding to site i on day j; $CWV_{ij}$ is the daily average CWV value (cm) corresponding to site i on day j; $POP_i$ is the population over the area covering the site *i*; $\alpha$ and $\mu_j$ are the fixed and random intercepts on day j, respectively; $\beta$ and $\upsilon_j$ are the fixed and random slopes for AOT on day j, respectively; $\gamma$ and $\varphi_j$ are the fixed and random slopes for CWV on day j, respectively; $\varepsilon_{ij}$ is the error term at site i on day j. In this model, the fixed parameter $\beta$ and $\gamma$ of AOT and CWV represent the average effects on $PM_{2.5}$ concentrations for the entire space and time in the study area, while the random parameter $\upsilon_j$ and $\varphi_j$ (day-specific) explain daily variation of the $PM_{2.5}$-AOT relationship. The above statistical model (Equation (5)) was applied to the entire sample pairs of $PM_{2.5}$–AOT to calculate the fixed-effect intercept and slope for all the model-valid days and random effect intercepts and slopes (day-specific) for each individual model-valid day. Finally, we get $PM_{2.5}$

estimation model for each model-valid day. The model coefficients were calculated using package 'lme4' of R programming language (version 3.4.3).

For model validation, 10-fold cross-validation (CV) [65] was applied to test the model potential over-fitting through randomly dividing the entire sample pairs (i.e., $PM_{2.5}$–AOT pair) into 10 subsets with almost 10% of the whole sample pairs for each subset. The process of CV was conducted through using nine sample subsets for model fitting, while the remaining one sample subset for model testing, then repeated 10 times until every subset was tested and combined all the testing subsets. We assessed the model performance by calculating the overall $R^2$, the site-specific (each site) $R^2$ between $PM_{2.5}$ estimates and ground ground-based measurements, mean prediction error (MPE), and root mean squared prediction error (RMSPE) for model fitting and CV results.

$$MPE = \sum_{i=1}^{n}\left(y'_i - y_i\right)/n \tag{6}$$

$$RMSPE = \sqrt{\sum_{i=1}^{n}\left(y'_i - y_i\right)^2/n} \tag{7}$$

where, $y'_i$ and $y_i$ are the estimated $PM_{2.5}$ and the measured $PM_{2.5}$ of the $i^{th}$ sample, respectively, and $n$ is the total number of $PM_{2.5}$-AOT pair samples.

In this paper, a model that estimates daily ground-level $PM_{2.5}$ concentrations was established. Using the model, we generated seasonal and yearly averaged $PM_{2.5}$ concentrations using the daily $PM_{2.5}$ estimates in the study area across the whole study period.

## 3. Results

### 3.1. Data Descriptive Statistics

In our study, in order to increase spatial coverage, the AOT average (daily AOT) of MAIAC Terra AOT (~ 10:00 local time) and MAIAC Aqua AOT (~ 14:00 local time) were used to establish $PM_{2.5}$ estimation models. We built the linear regression models for warm and cold seasons based on days where both AOT in the morning and AOT in the afternoon are available, then missing AOTs were estimated via above linear models. After adjusting, the size of model fitting dataset was dramatically increased, with $PM_{2.5}$–AOT sample pairs increased from 529 to 1635 and corresponding model-valid days from 53 to 129 during 2015. Data statistics summary shows that the overall mean of $PM_{2.5}$ 24-h average is 58.25 μg/m³, varying from 29.88 μg/m³ (SD = 15.60 μg/m³) to 82.87μg/m³ (SD = 48.28 μg/m³) by site (see Supplementary: Table S2) and the overall mean of $PM_{2.5}$ time average of 10:00 and 14:00 is 60.70 μg/m³, varying from 23.86 μg/m³ (SD = 13.24 μg/m³) to 88.35 μg/m³ (SD = 57.3 μg/m³) by site (see Supplementary: Table S3); the overall mean MAIAC AOT is 0.64, varying from 0.39 (SD = 0. 20) to 0.79 (SD = 0.38) by site (Table S4). Figure 1c shows that high $PM_{2.5}$ level mainly occurs at Chengdu, Deyang, and Meishan, located in the central and south areas of the study area, which might likely be due to high population density, vehicles, climate (high humidity all year), and terrain condition; the MAIAC AOT average (Figure 1d) and $PM_{2.5}$ levels for the entire study period (Figure 1d) have a similar spatial pattern.

### 3.2. Results of Model Fitting and Validation

In our study, we established several models for daily estimation of ground-level $PM_{2.5}$ with different $PM_{2.5}$ time average datasets and AOT datasets. Firstly, $PM_{2.5}$ estimation models were fitted using the MAIAC AOT average, $PM_{2.5}$ 24-h average (model-I), and $PM_{2.5}$ time average of 10:00 and 14:00 (model-II). Based on $PM_{2.5}$ estimates and ground-based measurements, we calculated $R^2$, intercept, slope, RMSPE, and MPE between them for each monitoring site. Figure 2a shows the boxplots statistics of model fitting performance at all the 36 sites, and site-specific $R^2$ ranges from 0.60

to 0.96, slope is from 0.68 to 1.20, RMSPE is from 9.08 μg/m$^3$ to 26.36 μg/m$^3$, MPE is from 6.86 μg/m$^3$ to 22.57 μg/m$^3$ for Model-I; and R$^2$ ranges from 0.46 to 0.94, slope is from 0.65 to 1.10, RMSPE is from 11.76 μg/m$^3$ to 33.22 μg/m$^3$, MPE is from 8.40 μg/m$^3$ to 22.31 μg/m$^3$ for Model-II; and median values of R$^2$, slope for Model-I (0.84) is higher than corresponding values of Model-II (0.78), while RMSPE has an adverse result (14.88 μg/m$^3$ versus 19.24 μg/m$^3$), indicating that Model-I has much better estimation performance than Model-II, and good agreement between fitted and measured PM$_{2.5}$ at each monitoring site. The overall R$^2$ of all the sites is 0.81 for Model-I, while it is 0.78 for Model-II. The scatterplots of PM$_{2.5}$ estimates of model fitting versus ground-based PM$_{2.5}$ measurements can be seen in Figure 3a. Table 1 shows the mode performance of Model-I and Model-II, demonstrating that both PM$_{2.5}$ 24-h average and PM$_{2.5}$ time average of 10:00 and 14:00 can be used to fit estimation models with good estimation performance, and among them, model-fitted ith PM$_{2.5}$ 24-h average is much better. In addition, model CV performance statistics results are shown in Figures 2b and 3b; the performance statistics of model CV (Figure 2b) shows that site-specific R$^2$ for both Model-I and Model-II decrease, while RMSPE and MPE increase from model fitting to model validation with small differences. On the other hand, model CV results shown in Figure 3b and Table 1 indicate that overall R$^2$ of Model-I and Model-II are 0.77 and 0.74, respectively, for the entire study period. Both Model-I and Model-II have over-fitting due to R$^2$ decrease and RMSPE and MPE increase, yet the differences are small. Overall, although both Model-I and Model-I have good estimation performance with higher R$^2$ values, Model-I is better due to lower RMSPE and MPE. So, we employed Model-I to estimate daily ground-level PM$_{2.5}$ concentrations over the study area for all model-valid days in the year of 2015.

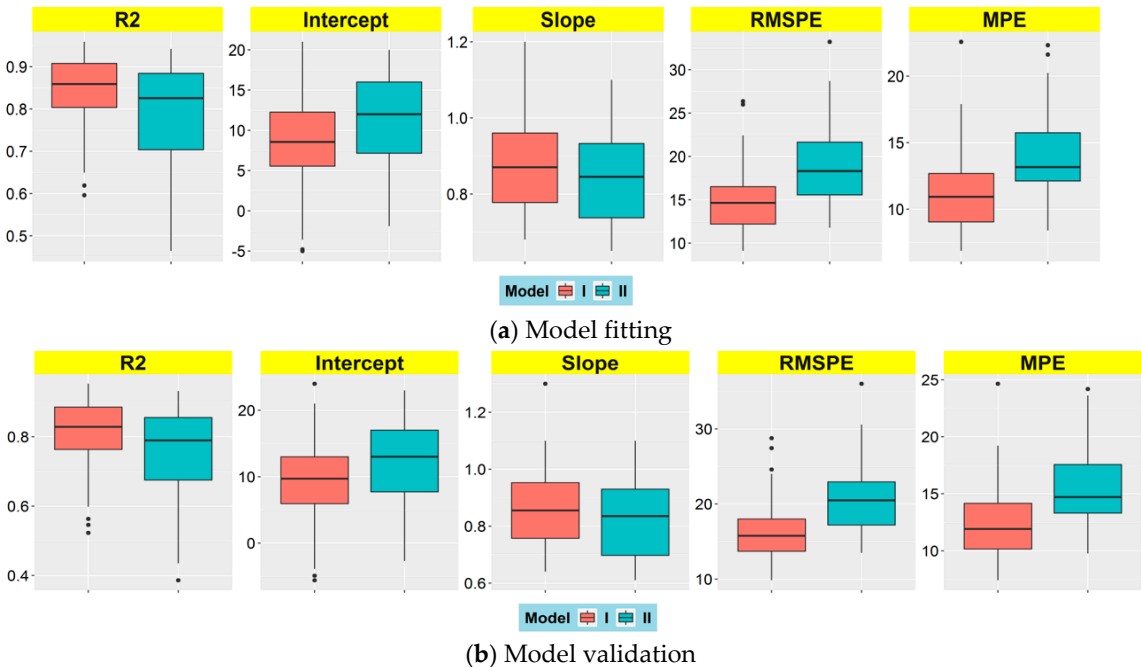

**Figure 2.** Boxplots statistics of estimation performance at all the 36 sites for R$^2$, intercept, slope, root mean squared prediction error (RMSPE) (μg/m$^3$), MPE (μg/m$^3$). (**a**) Model fitting; (**b**) model validation. 'I' denotes models using PM$_{2.5}$ 24-h average, and 'II' denotes models using PM$_{2.5}$ time average of 10:00 and 14:00.

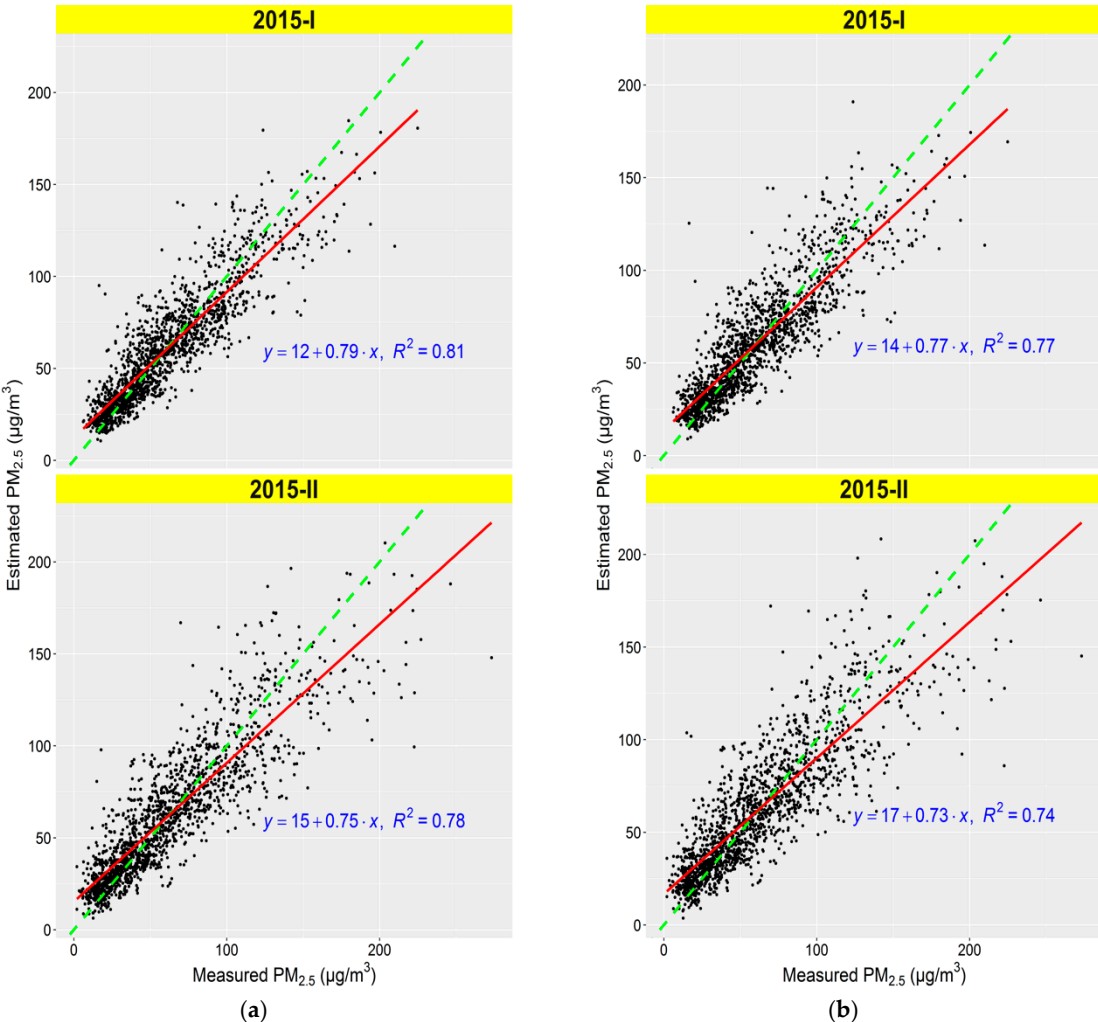

**Figure 3.** Scatterplots for model fitting and cross validation between estimated and measured PM$_{2.5}$ for each year. (**a**) Model fitting; (**b**) model cross validation. The red solid line represents the regression line and the green dashed line is the 1:1 line with slope of 1.0; 'I' denotes models using PM$_{2.5}$ 24-h average, and 'II' denotes models using PM$_{2.5}$ time average of 10:00 and 14:00.

**Table 1.** Model performance summary.

| Model | N [1] | M [2] | Model Fitting | | | Cross Validation | | |
|---|---|---|---|---|---|---|---|---|
| | | | R$^2$ | RMSPE (µg/m$^3$) | MPE (µg/m$^3$) | R$^2$ | RMSPE (µg/m$^3$) | MPE (µg/m$^3$) |
| Model-I [3] | 129 | 1635 | 0.81 | 15.47 | 11.09 | 0.77 | 17.04 | 12.31 |
| Model-II [4] | 129 | 1635 | 0.78 | 19.96 | 14.09 | 0.74 | 21.78 | 15.53 |
| Model-III [5] | 110 | 1346 | 0.65 | 27.03 | 19.57 | 0.58 | 29.33 | 21.42 |
| Model-IV [6] | 80 | 762 | 0.85 | 17.84 | 11.90 | 0.80 | 20.18 | 13.64 |
| Model-V [7] | 53 | 529 | 0.86 | 14.62 | 10.56 | 0.82 | 16.56 | 12.11 |
| Model-VI [8] | 53 | 529 | 0.82 | 19.76 | 13.85 | 0.78 | 22.12 | 15.63 |

[1] Denotes the number of days used in model fitting. [2] Denotes the number of PM$_{2.5}$-AOT pairs used in model fitting. [3] Denotes models using PM$_{2.5}$ 24-h average and AOT average of adjusted Terra and Aqua MAIAC AOT. [4] Denotes models using PM$_{2.5}$ average between 10:00 and14:00, and AOT average of adjusted Terra and Aqua MAIAC AOT. [5] Denotes models using Terra MAIAC AOT and average between 10:00 and 11:00. [6] Denotes models using Aqua MAIAC AOT and PM2.5 average between 13:00 and 14:00. [7] Denotes models using PM$_{2.5}$ 24-h average and AOT average of Terra and Aqua MAIAC AOT. [8] Denotes models using PM$_{2.5}$ average between 10:00 and 14:00, and AOT average of Terra and Aqua.

In addition, models are also established using just one satellite observation (Table 1), i.e., Model-III fitted with Terra AOT and PM$_{2.5}$ time average of 10:00 and 11:00, Model-IV fitted with Aqua AOT and PM$_{2.5}$ time average of 13:00 and 14:00, Model-V fitted with the average of MAIAC Terra AOT and MAIAC Aqua AOT (both of them are available in a given day) and PM$_{2.5}$ 24-h average, and Model-VI fitted with the average of MAIAC Terra AOT and MAIAC Aqua AOT (both of them are available in a given day) and PM$_{2.5}$ time average of 10:00 and 14:00. It should be noted that the missing AOT is not estimated for Model-V/VI. Model estimation performance of above four models are also given in Table 1, indicating that the model fitted R$^2$ are higher for most models except Mode-III, and Model-V with the highest R$^2$, and lowest RMSPE and MPE. Models fitted with PM$_{2.5}$ 24-h average (Model-I, Model-V) are better than those with PM$_{2.5}$ time average of 10:00 and 14:00 (Model-II, Model-VI) due to higher R$^2$ and lower RMSPE and MPE. Also, the model fitted with MAIAC Aqua AOT (Model-IV) is better than that with MAIAC Terra AOT (Model-III), with higher R$^2$ and lower RMSPE and MPE but with less model-valid days and PM$_{2.5}$–AOT sample pairs.

The differences between Model-I estimated and ground-based PM$_{2.5}$ concentrations are shown in Figure 4. It can be seen that PM$_{2.5}$ concentrations are slightly overestimated in lightly polluted areas, while underestimated at heavily polluted areas. Nevertheless, the difference were within ± 17.13 μg/m$^3$ for 80% of all the monitoring observations.

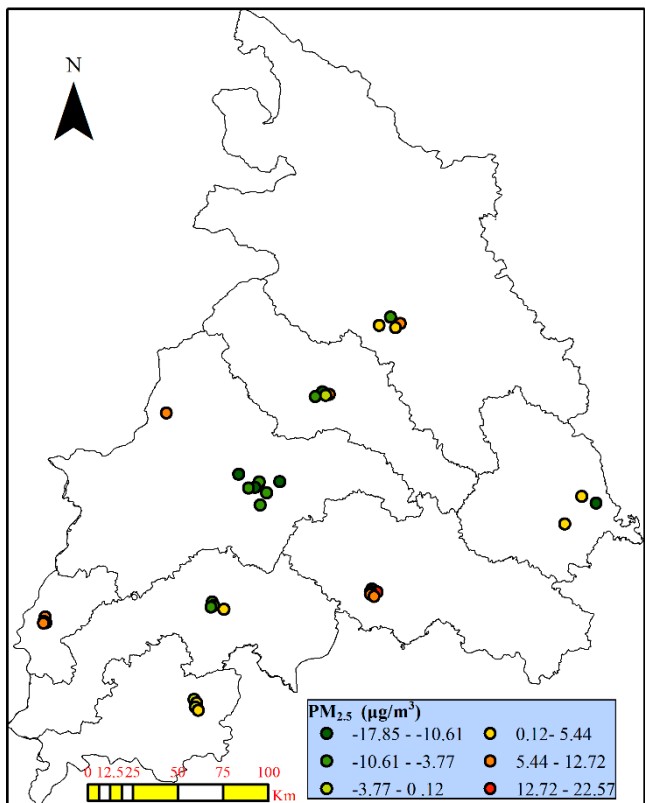

**Figure 4.** The differences between estimated and measured PM$_{2.5}$ (μg/m$^3$) at each monitoring site during 2015.

### 3.3. Spatiotemporal Trends of PM$_{2.5}$ Concentrations

We employed the Model-I to estimate daily PM$_{2.5}$ concentrations, then the maps of weekly, monthly, seasonal, and annual mean PM$_{2.5}$ can be created, which are vital for the prevent and control of air pollution for local government or travel and life of city residents. As discussed before, there are a total of 129 model-valid days that have daily PM$_{2.5}$ estimates during the entire study period. Figure 5a illustrates the spatial distribution of PM$_{2.5}$ average from all the model-valid days at 1 km resolution for the entire study area (i.e., urban agglomeration of Chengdu Plain). It can be observed that the

PM$_{2.5}$ levels have obvious spatial change pattern across the study area: High PM$_{2.5}$ levels are mainly located in the middle and southern areas, which are the mountain valleys of Longquan Mountains and Qionglai Mountians (Figure 5a); while low PM$_{2.5}$ levels appear in rural and mountainous areas in the northwest and southwest areas. Such spatial patterns correspond well with those of ground-based monitoring sites shown in Figure 1b. It is evident that the boundaries of plains and mountains also separate areas of high PM$_{2.5}$ levels areas and low PM$_{2.5}$ levels areas over Chengdu Plain, indicating that the terrain has significant impact on the distribution of pollution.

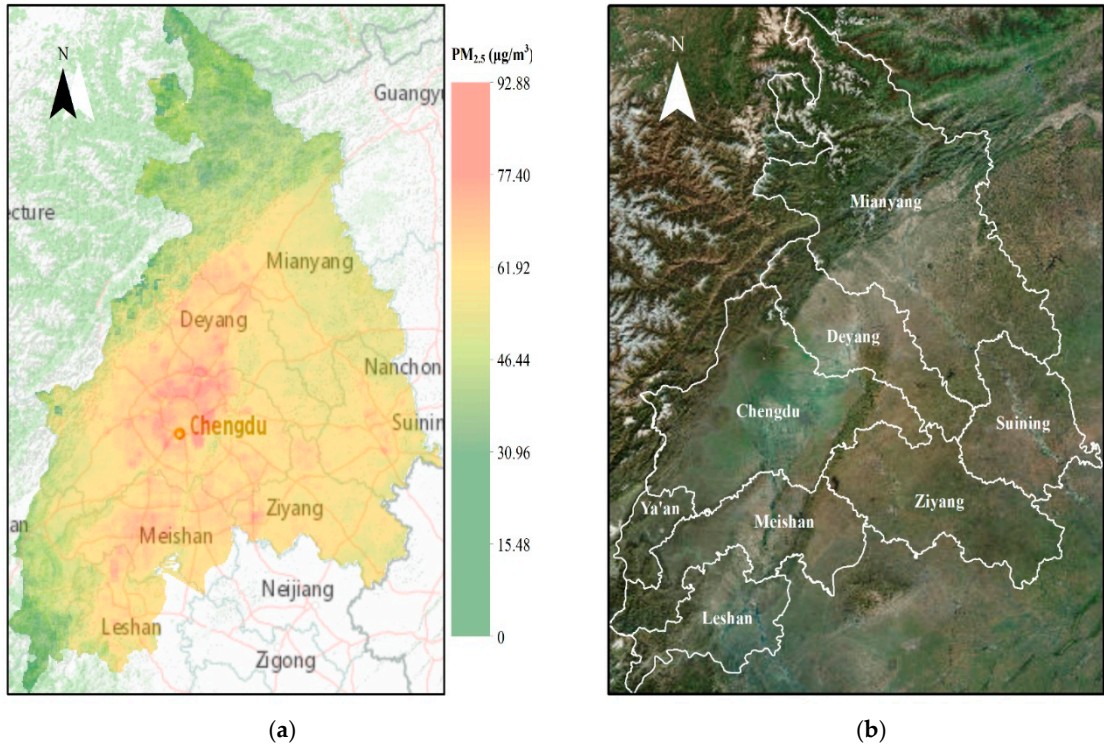

(**a**)                    (**b**)

**Figure 5.** (**a**) Maps of mean PM$_{2.5}$ concentration ($\mu$g/m$^3$) averaged from all PM$_{2.5}$ estimates model valid days for 2015, and the back ground is traffic map. (**b**) Satellite imagery map for Chengdu Plain (Satellite imagery source: Esri, DigitalGlobe, GeoEye).

The PM$_{2.5}$ average is 56.86 $\mu$g/m$^3$ during the whole study period, which is comparable to 53.86 $\mu$g/m$^3$ of all the monitoring sites, 1.5 times more than China NAAQS of 35.0 $\mu$g/m$^3$ for the daily mean PM$_{2.5}$. The city-specific mean PM$_{2.5}$ was the average of all PM$_{2.5}$ grid cells that belong to one specific city, varying from 49.78 $\mu$g/m$^3$ for Ya'an to 62.12 $\mu$g/m$^3$ for Chengdu. The PM$_{2.5}$ estimates indicate that almost the majority of people living in the study area are exposed to risky PM$_{2.5}$ pollution levels, especially Chengdu, the provincial capital of Sichuan Province, with much dense population, which is a serious public concern for local government and common people. Figure 5a indicates that high PM$_{2.5}$ level areas are located at areas with high-level urbanization in Chegndu Plain. Figure 5a also indicates that higher PM$_{2.5}$ levels are located at main traffic roads, and several atmospheric pollution hotspots exist at a traffic hub with high density of roads in Chengdu, indicating that traffic vehicles are an important source of PM$_{2.5}$, also high PM$_{2.5}$ levels in urban center of Suining, Meishan, Ziyang, and Leshan.

Moreover, the seasonal mean PM$_{2.5}$ estimates in the whole study area are 49.80, 30.16, 48.61, and 88.45 $\mu$g/m$^3$ for spring (March, April, and May), summer (June, July, and August), autumn (September, October, and November), and winter (December, January, and February), respectively. From Figure 6a–d, it is easily observed that the PM$_{2.5}$ levels has a remarkable seasonal variability, with the highest PM$_{2.5}$ level during the winter and the lowest during the summer. The high PM$_{2.5}$

concentrations during the winter are possibly caused by enhanced anthropogenic emissions from fossil fuel combustion and biomass burning, as well as unfavorable meteorological conditions such as temperature inversion during the cold periods or lower mixing height for atmospheric pollutants dispersion; the low $PM_{2.5}$ concentrations during the summer are possibly due to reduced anthropogenic emissions, high humidity, and more rainfall, which is helpful for depositions of aerosol particles.

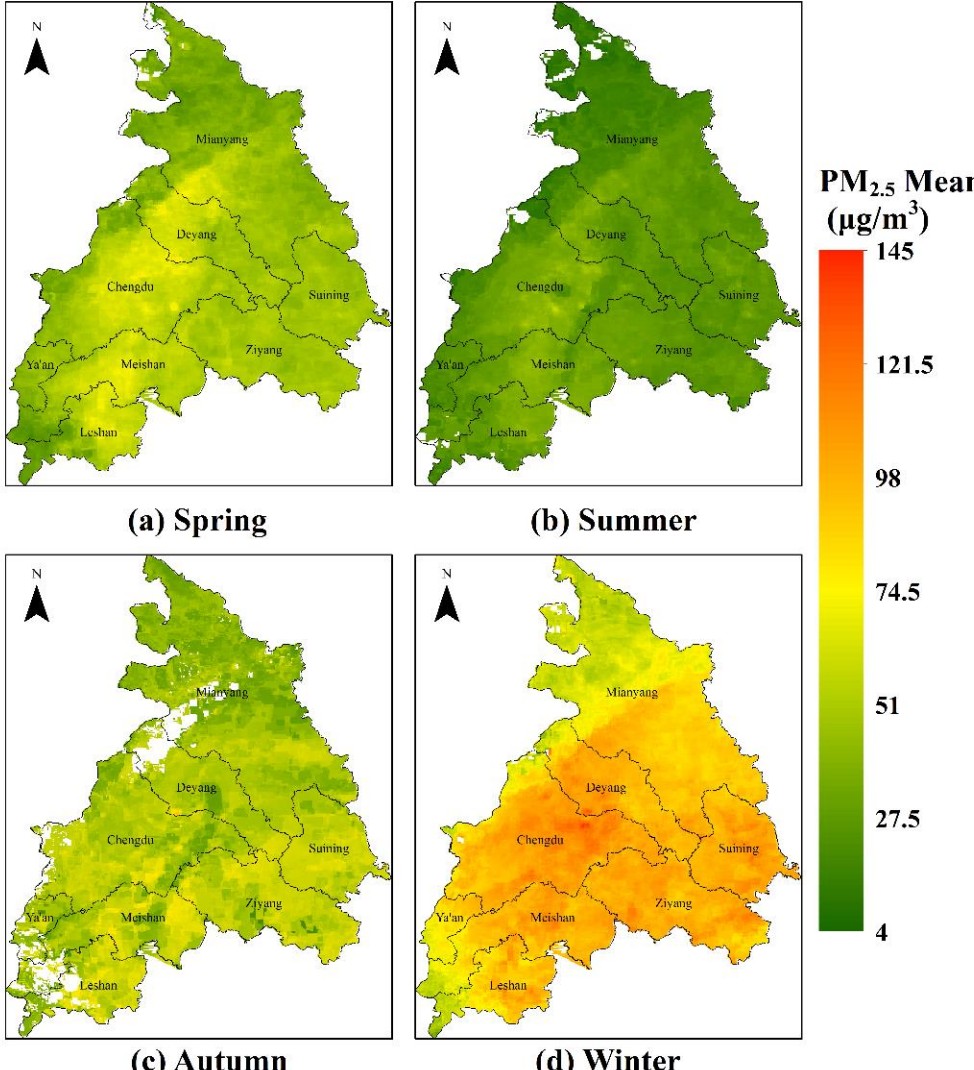

**Figure 6.** Maps of seasonal mean $PM_{2.5}$ concentration ($\mu g/m^3$) calculated with daily estimation for the year of 2015. (**a**) Spring, (**b**) summer, (**c**) autumn, (**d**) winter. The white blank areas indicate missing value due to lack of MAIAC AOT data. The boundaries of eight cities area shown in black lines.

## 4. Discussion

In our study, we established and compared several daily ground-level $PM_{2.5}$ estimation models using different $PM_{2.5}$ datasets, AOT datasets, and other auxiliary variables. The results show that both model fitting and CV for all models can generate higher $R^2$ (Table 1). Among the models, the Model-I has the best performance and agreement between satellite estimated and ground-based measured $PM_{2.5}$, with model-valid days of 129 and $PM_{2.5}$–AOT pairs of 1635. Compared to linear regression model performance based on the same modeling data (see Supplementary: Figure S2), the performance of Model-I is hugely improved when considering daily variations of $PM_{2.5}$–AOT relationship. Model-CV results show models have over-fitting due to $R^2$ decrease, RMSPE, and MPE increase. Model-I/Model-V has slightly better performance than Model-II/Model-VI shown in Table 1.

For models fitted with only one satellite observation (Terra or Aqua), Model-III (Terra AOT, morning) is inferior to Model-IV (Aqua AOT, afternoon), with higher RMSPE and MPE, and lower $R^2$. Besides, after estimating the missing AOT data, the number of $PM_{2.5}$–AOT pairs was significantly increased from 529 to 1635, and also improved spatial and temporal coverage of $PM_{2.5}$ estimates, only causing a slight degradation of performance. Thus, the process of estimating of the missing AOT is important and necessary for the study area with serious data missing.

The air pollution strongly depends on thermodynamic state of the atmosphere. The turbulence occurring in the boundary layer of the atmosphere influences the meteorological conditions, and especially the convection process and temperature inversion. It strongly influences the formation and intensity of diffusion in the atmosphere and, as a consequence, the concentration of primary and secondary atmospheric pollutants in the lower troposphere. For one day, after sunrise, the radiation increases, and the mixing layer height increases rapidly under the impact of thermal buoyancy lift, reaching the maximum at about midday. Meanwhile, the atmospheric pollutants emitted at ground will be lifted aloft along with mixing layer height rising. After noon, the mixing layer height decreases as a consequence of the reduction of convective activity driven by the heating of the Earth' surface by sunlight and the corresponding nocturnal radiative cooling of the ground. Atmospheric pollutants (e.g., particles) confined within the mixing layer will descend to the ground due to inactive of thermodynamic state and gravitational force. Hence, the concentration of atmosphere particles increases [66].

Satellite-derived AOT represents the amount of aerosol in the whole vertical column of atmosphere from the ground to satellite altitude (not only limited to the vertical column segment close to the ground-based monitoring sites). According to the varying pattern of the atmospheric pollutants in one day described above, the $PM_{2.5}$ 24-h average captures more of the aerosols subsiding down to ground level at monitoring sites than does the $PM_{2.5}$ time average of 10:00 and 14:00. This might explain why the Model I/V performs a little better than the Model II/VI.

As described above, the atmospheric pollutants are confined within the mixing layer, and lifted to the atmosphere. The atmospheric pollutants during satellite Aqua overpass are more uniformly mixed within the mixing layer than those during satellite Terra overpass, so the correlation between $PM_{2.5}$ and AOT is much stronger, which is likely to interpret that performance of Model-IV using MAIAC Aqua AOT is better than Model-III using MAIAC Terra AOT.

In our study, the Model-I gained higher model fitting $R^2$ of 0.81 and CV $R^2$ of 0.77 than previous studies, e.g., the GWR model applied to the whole China mainland with overall CV $R^2$ of 0.64 [30], to PRD area with $R^2$ of 0.74 [33]; VIIRS night light data incorporated and applied to BTH region with $R^2$ of 0.75 [67]; an improved model applied to three megalopolises with high pollution in China, namely, to the BTH with $R^2$ of 0.77, to the YRD region with $R^2$ of 0.80, and to the PRD region with $R^2$ of 0.80 [43]; the observation-based method considered the effect of the main aerosol particle characteristics applied to BTH region, YRD, and PRD, with $R^2$ of 0.70 (N = 331), 0.77 (N = 279), and 0.83 (N = 329) [47]; the LMEM models using only MODIS AOT at 3 km spatial resolution were applied to Beijing with $R^2$ of 0.796 and 0.81 [44,45]. The similar models were applied in US with MAIAC AOT, e.g., applied to southeastern US with model fitting $R^2$ of 0.83 and CV $R^2$ of 0.67 [35], long-term (10 years) analysis with model fitting $R^2$ of 0.71–0.85 and CV $R^2$ of 0.62–0.78 [36] and also in northeastern US with CV $R^2$ of 0.89 [68]. The RMSPE of Model-I CV was 17.04 $\mu g/m^3$, which was higher than than in the US (<9.0 $\mu g/m^3$) [21,29,34–38], but comparable to Beijing area with a coarser 3 km spatial resolution of $PM_{2.5}$ estimates by Li (16.04 $\mu g/m^3$) [44] and Xie (17.85 $\mu g/m^3$) [45] but with higher $R^2$, possibly due to the smoothing effect of satellite data with coarser resolution, representing more coverage condition of air quality. The higher RMSPE in our study area or other Chinese areas than the US was more likely due to much higher $PM_{2.5}$ levels, ranging from 6.0 to 224.83 $\mu g/m^3$ with the average of 59.33 $\mu g/m^3$, which was up to several times as that in the United States.

Although the satellite-derived $PM_{2.5}$ can provide larger spatial coverage than ground-based monitoring sites, the satellite has less temporal coverage due to its observation limitations, e.g., clouds, fogginess, surface conditions, and other factors. In our study, there are a total of 129 model-valid

days, which is only about 35% for one year. In addition, due to cloudy or foggy weather in Chengdu Plain causing low sampling frequency of available satellite observations, the larger variations in the $PM_{2.5}$ levels were possibly be neglected. Satellite-derived AOT should be used carefully when estimating ground-level $PM_{2.5}$ concentrations in Chengdu Plain due to the data missing in heavy pollution conditions where the AOT retrieval algorithm is not valid. As air pollution has become an increasing concern for the public and government, the ground-based monitoring sites will be set up more and more, and further studies should be conducted by considering other factors affecting the $PM_{2.5}$–AOT relationship.

Overall, the model-estimated $PM_{2.5}$ concentrations have a good consistency with that of ground-based measurements. However, there are days and sites with large residuals as seen in Figure 3. We tried to get more accurate $PM_{2.5}$ estimation at every place and every day, but there are many factors that impact the spatial variability of $PM_{2.5}$-AOT relationship; in particular, mixing height, $PM_{2.5}$ composition, temperature, and humidity [50,51]. Another reason is the number of ground monitoring $PM_{2.5}$ sites and its uneven spatial location.

## 5. Conclusions

In this study, we developed an improved linear mixed effects model (LMEM) to estimate the daily ground-level $PM_{2.5}$ concentrations referring to atmospheric particulate matter with a diameter of less than 2.5 μm by incorporating humidity and gridded population data. It is should be noted that to reduce the uncertainty caused by inconsistency of spatiotemporal between aerosol optical thickness (AOT) and meteorological data, both of them should be generated from the same satellite.

The model is implemented to the urban agglomeration of Chengdu Plain, where heavy atmospheric particle pollution is common and frequent. The results indicate that the $PM_{2.5}$–AOT relationship was significantly improved when considering the day-to-day variability, with a much higher accuracy than simple linear regression (see Supplementary: Figure S3) with $R^2$ of 0.81 versus 0.49, lower root mean squared prediction error (RMSPE) of 15.47 μg/m$^3$ versus 25.32 μg/m$^3$, mean prediction error (MPE) of 11.09 μg/m$^3$ versus 18.87 μg/m$^3$; the mean $PM_{2.5}$ estimates across the whole study area was 56.86 μg/m$^3$, which is comparable to 53.86 μg/m$^3$ of all the monitoring sites, indicating good consistency of $PM_{2.5}$ estimates and ground monitoring sites.

The results demonstrate that the $PM_{2.5}$ levels in the study area have an obvious spatial pattern, with high $PM_{2.5}$ levels mainly located in middle and southern areas, and low $PM_{2.5}$ levels in rural and mountainous areas. The $PM_{2.5}$ levels also have a remarkable seasonal variability, with the highest level during the winter and the lowest during the summer.

Besides, the product of daily $PM_{2.5}$ estimation at 1 km spatial resolution in heavily polluted areas with high accuracy are valuable for local government pollution monitoring, public health research, and urban air quality control.

**Supplementary Materials:** The following are available online at http://www.mdpi.com/2073-4433/10/5/245/s1, Figure S1: Scatterplots of MAIAC AOT versus AERONET, Figure S2: Scatterplots of daily $PM_{2.5}$ versus daily AOT, Figure S3: Scatterplots of Estimated $PM_{2.5}$ versus Measured $PM_{2.5}$, Table S1: Geolocation of 36 Monitoring Sites in Urban Agglomeration of Chengdu Plain, Table S2: Statistical Summary of Daily $PM_{2.5}$ 24-h Average for Each Site in Urban Agglomeration of Chengdu Plain, Table S3: Statistical Summary of Daily $PM_{2.5}$ time average of 10 a.m. and 2 p.m. for Each Site in Urban Agglomeration of Chengdu Plain, Table S4: Statistical Summary of Daily AOT Average for Each Site in Urban Agglomeration of Chengdu Plain.

**Author Contributions:** W.H., conceived and designed the experiments; W.H. and L.T. analyzed the data; W.H. wrote the original draft of the paper, which was revised by L.T.

**Funding:** This research was funded by the National Natural Science Foundation of China (Grant No. 41771433) and Advanced Research Project (Grant No. 30102060301).

**Acknowledgments:** The authors would like to thank the MAIAC algorithm team for offering AOT data, SEDAC team for offering GPWv4 data.

**Conflicts of Interest:** The authors declare no conflict of interest.

## Appendix A

**Table A1.** The acronyms and definitions used in this article area displayed below table.

| Acronyms | Definition |
|---|---|
| AERONET | Aerosol Robotic Network |
| AOT | Aerosol Optical Thickness |
| AVHRR | Advanced Very High Resolution Radiometer |
| BRDF | Bidirectional Reflectance Distribution Function |
| BTH | Beijing-Tianjin-Hebei |
| CV | Cross-Validation |
| CWV | Column Water Vapor |
| GAM | Generalized Additive Model |
| GPW | Gridded Population of the World |
| GPWv4 | GPW collection in fourth version |
| GWR | Geographically Weighted Regression |
| LMEM | Linear Mixed Effect Model |
| LUR | Land Use Regression |
| MAIAC | Multi-Angle Implementation of Atmospheric Correction |
| MAIAC AOT | AOT products produced with MAIAC algorithm |
| MAIAC CWV | CWV products produced with MAIAC algorithm |
| MAIAC-Aqua AOT | AOT products produced with Aqua satellite using MAIAC algorithm |
| MAIAC-Terra AOT | AOT products produced with Terra satellite using MAIAC algorithm |
| MISR | Multi-angle Imaging SpectroRadiometer |
| MOD04 | MODIS Aerosol Product with 10 km resolution |
| MODIS | MODerate resolution Imaging Spectroradiometer |
| MPE | Mean Prediction Error |
| China NAAQS | China National Ambient Air Quality Standard |
| NASA | National Aeronautics and Space Administration |
| $PM_{2.5}$ | Particulates with aerodynamic diameters of less than 2.5 μm |
| POP | Population data |
| PRD | Pearl River Delta |
| RMSPE | Root Mean Squared Prediction Error |
| SeaWiFS | Sea-Viewing Wide Field-of-View Sensor |
| SEDAC | Socioeconomic Data and Application Center |
| SR | Surface Reflectance |
| SRC | Spectral Regression Coefficient |
| TEOM | Tapered Element Oscillating Microbalance |
| TOMS | Total Ozone Mapping Spectrometer |
| VIIRS | Visible Infrared Imaging Radiometer Suite |
| WHO | World Health Organization |
| YRD | Yangtze River Delta |

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
