# Peer review of "Satellite-Based Estimation of Daily Ground-Level PM2.5 Concentrations over Urban Agglomeration of Chengdu Plain"

_atmosphere, doi:10.3390/atmos10050245_

Round 1

Reviewer 1 Report

In the paper, “Satellite-Based Estimation of Daily Ground-Level PM2.5 Concentrations over Urban Agglomeration of Chengdu Plain”, the authors derive daily PM2.5 concentrations over the Chengdu Plain for a one-year (2015) period using satellite/population/surface network datasets.  While the authors have completed a great deal of work, and have also provided supplementary material, I have several major concerns with the submitted manuscript.  It needs a substantial edit prior to being acceptable for publication.  Thus, I recommend major revisions for this paper. The authors should address the following list of major and minor comments for the revised manuscript.   

Major comments:

1.     Grammar/spelling/narrative: The entire manuscript is filled with grammatical errors, including misspellings, incomplete/incorrect acronyms, and incorrect sentence structure.  This makes the paper very difficult to read, and alone warrants a major revision.  I strongly suggest that the manuscript be carefully proofread by a native English reader to improve the narrative.     

2.     Originality of the study: There have been many studies in the past 10-15 years during which the investigators have attempted to estimate surface PM2.5 from satellite AOT data.  Some have been simple correlative studies, while others have used model fitting with various parameters to improve the PM2.5/AOT relationship (like this study).  Since similar research has been done before, the authors need to give reasons as to why this manuscript provides new information/knowledge of this topic. Please add this to the end of the introduction section.         

3.     The aerosol vertical distribution issue: The authors correctly point out that the PM2.5/AOT relationship is affected by the distribution of aerosols within the atmospheric profile.  This is because AOT is a column-integrated value and PM2.5 represents a measurement near the surface.  Thus, any elevated aerosol layers will negatively affect the relationship. However, Equation 1 does not contain a term to account for the impact of this issue.  Why is this so?  The authors need to account for this in their study.  Also, please add some more discussion to the introduction of past work that has been done concerning the impacts of aerosol vertical distribution when estimating surface PM2.5 from passive satellites (using different lidars, etc.).  For example, Toth et al. (2014) examined this topic using observations from CALIOP. 

Minor comments:

1.     Page 1, Line 28: Define “PM2.5”.  Also, define all other acronyms throughout the manuscript. For example, in the Introduction section alone: AOT, AVHRR, TOMS, MODIS, MISR, and SeaWIFS all must be defined. Please do this for other sections as well.

2.     Page 3, Lines 93-102: Please add a few details of the uncertainties expected in the ground-based PM2.5 measurements.   

3.     Page 3, Line 105: Add more explanation to what is meant by “h02v02 and h03v02”.  Also, was it meant by “tiles”?

4.     Page 5: I suggest adding latitudes and longitudes to the borders of each map in Figure 1.

5.     Page 5: For Figure 1b, please state what satellite the image is taken from.

6.     Page 5: For Figure 1d, the color bar should be reversed (the largest numbers should be on the right not the left).  If you still want the red color to represent large AOTs, please reverse the colors as well.

7.     Page 7, Line 237 (and caption for Figure 2): Are you sure that you mean “at each site”?  It seems to me that this represents a statistical summary of all sites analyzed.  Is that correct?

8.     Page 7, Line 245: Table 1 is mentioned later in the text, but appears earlier in the paper.  For example, Figure 2 is discussed before Table 1, yet it appears after Table 1 in the paper.  I suggest fixing this in the revision by switching Table 1 and Figure 2 on Pages 6 and 7.

9.     Page 8: For Figure 3, are the regression lines from simple linear regression?  If so, I suggest replacing these lines with those computed from Deming regression analysis (https://en.wikipedia.org/wiki/Deming_regression).  This is because it considers errors in both the x and y axes, making it applicable for PM2.5 studies.    

10.  Page 9, Lines 285-288: So, you are relating the satellite visible image in Figure 5b to PM2.5 presence near the surface?  In the image, it is difficult to see the aerosols. I would better clarify in the text or find another way to illustrate this point.

11.  Page 10 (Figure 4): This is not a very informative figure as currently presented.  I recommend removing it or heavily revising it.  In the blue box in the lower right-hand corner, why are there two numbers for each green dot?  Why not just include the differences?   

12.  Page 11 (Figure 5): In Figure 5a, the color bar should be reversed, with the largest values on the right.  The colors would then need to reversed as well, if you wish. The label for Figure 5b is incorrect, as it is currently labeled “a”.  Please change this.  Also, in the caption for Figure 5, state which satellite the image was taken from. 

13.  Page 12, Line 343: I think you may be missing a citation here for the VIIRS nighttime study you are referencing.  

14.  In the Conclusions section, all acronyms should be redefined. 

Paper cited:

Toth, T. D., Zhang, J., Campbell, J. R., Hyer, E. J., Reid, J. S., Shi, Y., and Westphal, D. L.: Impact of data quality and surface-to-column representativeness on the PM2.5 / satellite AOD relationship for the contiguous United States, Atmos. Chem. Phys., 14, 6049-6062, https://doi.org/10.5194/acp-14-6049-2014, 2014.

Reviewer 2 Report

The manuscript is an interesting contribution to the research activity related to the estimate of particulate matter concentration near the ground from satellites. The work is well put into the context of the existing literature, and presents a novel simple approach to convert satellite aerosol optical depths to near-surface PM. However, the manuscript needs improvements and clarifications as detailed here below:

Significant figures: the use of digits should be revised in light of the uncertainties. Almost all data are presented with 1-2 decimal figures, but this is completely unrelated to the corresponding uncertainty/range. For example, the averages given at lines 222-224 are 58.25 ug/m3 with a standard deviation of 15.6 ug/m3, i.e. the "uncertainty" on the mean has LESS significant figures than the average! This is not meaningful. Please carefully revise the use of ALL reported numbers in text and tables thoughout the manuscript.

2. Please add a final table with acronyms used in the manuscript. For example, those given at line 109 are not defined anywhere.

3. In paragraph 125-147: It is explained that the orignal AOT products where scaled for the effect of hygroscopicity, but the procedure is really unclear. The results is given in the equations at lines 140-145, but it is not explained how these coefficients are calculated or illustrated. Please clarify.

4. Regarding point 3: it looks like the impact of the correction could be significant, thus some kind of evaluation on the final calculated PM2.5 would be desiderable.

5. eq. 1: this is the core of the inversion method. It is unclear how the "fixed" and the "random" coefficients are calculated (e.g. from all data vs. daily means?). Moreover, an illustration or reporting of the values would be desiderable for clarity. Indeed, it is not clear which variables are the most significant in the regression procedure.

6. Line 199: please add a reference for the CV method, for reproducibility.

7. section 3.3: the information given in table 2 and figure 4 is inconsistent. The mean values in the table are very similar, while in the figure the differences are much higher, but apparently are calculated from the same dataset. Please clarify.

8. Line 294 and related results: "population-weighted mean". This looks like a repeated application of population weighting, In eq. 1, indeed, there is already popoluation used as a predictand of PM2.5, thus it is unclear the role of this additional weighting here. This reinforces the need to address point 4. Please clarify.

Reviewer 3 Report

Please see the uploaded review report for details.

Round 2

Reviewer 1 Report

The authors have made substantial revisions to the paper, but I do not believe it is suitable for publication just yet.  My biggest remaining concern is the writing quality of the paper.  Many grammar issues are still found throughout the text, and are too numerous for me to point out each one.  Two quick examples: repeating the word “aerosol” on Page 2, Line 60 and misspelling “Cycle” on Page 2, Line 73.  There are many others.  I suggest a major revision, and have outlined my comments below.  Please consider these for the next version of the manuscript.  

1. Grammar issues are still present, with both the original and revised texts, including several misspelled words.  Due to this, I believe it unacceptable for publication until these are sufficiently fixed.

2.  I think it was a good idea to add an appendix, but all acronyms should still be defined in the text, including the introduction and conclusions sections.

3.   “PM2.5” should be defined in the abstract.

4.  Please add labels (a-d) to Figure 6.

5. Ideally, I would like to see the vertical information included, although I understand that it would be a significant task if the authors do not have experience with these data.  However, I strongly suggest that any future papers by the authors in this research area incorporate the use of lidar data (for aerosol vertical distribution purposes).

Reviewer 2 Report

The Authors mostly addressed my comments from the first round of review. The manuscript is now almost ready for publication, but I insist on the first point, which was not addressed, i.e. the use of significant digits. Results cannot be presented with all those decimal digits everywhere, they are unmeaningful: I suggest to estimate the SDs first with 2 digits, and then round the corresponding mean value accordingly.

Last point: please add the exact version, name and link to repository of MAIAC data, for reproducibilty. Since it is a relatively new product there a number of versions and data sources.

Reviewer 3 Report

Please see the attached file for details.

Round 3

Reviewer 1 Report

The authors have sufficiently responded to all of the comments from my previous review.  Some English proofreading is still required, but I believe this will take place during the copy-editing stage of the manuscript.  One final comment, however.  Please add the labels to the caption of Figure 6, and refer to them in the text of the manuscript as well.  I recommend a minor revision, after which I suggest the paper be accepted for publication.  

Author Response

We added the labels (a-d) to the caption of Figure 6 and referred to them in line 384 as follow: “Seen from Figure 6 (a), (b), (c) and (d), it is easily observed that the PM2.5 levels…”

Reviewer 3 Report

The authors have addressed all my comments and concerns. The manuscript is in a good shape now and ready to publish. I suggest to accept it in present form. 

Author Response

 Thanks for your review and good suggestions.